# Augmented Reality in Professional Training: A Review of the Literature from 2001 to 2020

**Xu Han, Ying Chen** 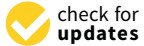**, Qinna Feng * and Heng Luo ***

Faculty of Artificial Intelligence in Education, Central China Normal University, Wuhan 430079, China; hanxu0513@mails.ccnu.edu.cn (X.H.); chenying2020@mails.ccnu.edu.cn (Y.C.)
* Correspondence: fengqinna@mails.ccnu.edu.cn (Q.F.); luoheng@mail.ccnu.edu.cn (H.L.)

**Abstract:** This study presents a systematic review of literature on the application of augmented reality (AR) in professional training contexts published between 2001 and 2020. A total of 49 articles were selected after a two-stage screening process, and key research findings were analyzed and synthesized using a coding scheme comprising five inter-related aspects: basic information, instructional contexts, technology features, instructional design, and research results. The review results depict the trend patterns in AR-supported professional training in terms of publication, research paradigm, and technological affordances, and report the contextual differences in AR pedagogies and instructional functions over time. Furthermore, a meta-analysis was conducted in the present study to examine the overall effectiveness of AR application in professional training, with the results indicating an overall small effect size ($g = 0.268$) and nine significant moderating factors. Informed by the review and meta-analysis results, a set of implications for facilitating and investigating AR-supported professional training are proposed and discussed.

**Keywords:** augmented reality; professional education; meta-analysis; systematic review; instructional design

## 1. Introduction

Augmented reality (AR) is defined as a technology-enhanced environment where virtual objects (augmented components) can be overlaid into the real world [1,2]. Azuma (1997) identified three technical features of AR: a combination of the real and virtual world, real-time interaction, and accurate 3D registration of virtual and real objects [1]. These AR features afford a highly immersive and interactive virtual experience for users, allowing them to observe, interact with, and create digitally enhanced reality individually or collectively. In recent years, the rapid technological advancement in smartphones and wearable devices has made AR technology more accessible and affordable, and thus kindled people's enthusiasm towards its usage in an educational context. However, while AR has been gradually adopted and increasingly investigated in K–16 education (K–12 and higher education), its application in professional training contexts has been scarce and exploratory, despite the importance of professional training for life-long learning and a knowledge-based society [3,4].

Professional training is defined as a set of behaviors and acts with the purpose of increasing the employees' professional skills to carry out a particular job in a better manner [5,6]. Such a definition highlights three important features of professional training. First, its purpose is educational, which focuses on employee development (e.g., skill acquisition and knowledge growth) rather than performance improvement. Second, its target learners are employees and professional staff instead of degree-seeking students. Third, it is job-specific and highly contextual, and thus often occurs in workplaces rather than traditional classroom settings. Professional training is commonly seen in the field of health and medicine, engineering, service sector, manufacturing industry, and teacher development.

For instance, continuing medical education is an educational form for resident doctors and interns who need to grow their practical skills in clinical diagnosis [7], and in-service teacher education is regularly implemented as part of the educator licensure requirement in many nations [8]. The typical forms of professional training include conference, lecture, workshop, and traineeship [9].

Professional training differs from K–16 education in its educational focus and pedagogy. While K–16 education focuses more on students' learning outcomes and academic achievements, professional education is more career-oriented, concerning the cost-effectiveness of the training program. Consequently, K–16 education often relies on the student-centered pedagogy to promote higher-order thinking and meta-cognitive skills [10], whereas professional training is more aligned with skill development featured by direct instruction and trial-and-error practice [11]. The literature highlights the importance of professional training as it has a positive impact on employees' working attitudes, job performance, and knowledge acquisition [6,12].

The aforementioned characteristics of professional training highlight the potential of AR as a proper instructional technology for this particular context. First, AR enables the natural integration of virtual instructional content into the actual working environment [13], which can promote situational cognition and experiential learning. For example, Abhari et al. (2015) described an authentic AR-enhanced surgery environment where novice physicians can improve their neurosurgical skills through hands-on practice [14]. Second, AR can provide a variety of visual cues in the digital forms of symbols, text, animation, or 3D objects, which are known to facilitate procedural learning in professional training [15]. Third, AR can facilitate a shared learning experience in groups owing to increased visibility of virtual content [16]. The ability to accommodate group learning can further improve the accessibility and feasibility of AR-supported professional training. Lastly, the digital artifacts afforded by AR allow for easy creation, modification, and duplication, which can greatly reduce the training cost for trial-and-error practice.

Recognizing the great potential of AR for developing professional competencies, this study systematically reviews the relevant literature published over the last two decades (2001–2020) with the purpose of extending our understanding of AR-supported instruction in professional training contexts. We synthesized the research findings through the lens of publication trend, application context, instructional design, and technical features, and determined the overall effectiveness of AR and its moderating factors through meta-analysis. Particularly, this study seeks to answer the following questions:

1. What are the trends of publications and research types for AR-supported professional training?
2. What are the essential technological features and affordances of AR that support professional training, and how are they evolving over time?
3. What instructional strategies have been employed in the AR-supported professional training?
4. What is the overall effectiveness of AR application in professional training and what are the moderating factors?

## 2. Methods

Following the standardized protocol proposed by Denyer and Tranfield [17], this study conducted a systematic review that involved two main stages: literature selection (including initial research search and manual screening) and analytical coding. Based on the research questions, we selected and coded literature of AR-supported professional training published between 1 January 2001 and 31 December 2020. In addition, a meta-analysis was conducted using the comprehensive meta-analysis (CMA) software (version 3) to examine the overall effectiveness of AR-supported instruction and its moderating factors.

*2.1. Literature Selection Process*

2.1.1. Initial Literature Search

An initial literature search was conducted in the literature database of Scopus (https://www.elsevier.com/en-in/solutions/scopus, accessed on 16 April 2021). Scopus is a widely used database for peer-reviewed literature and is recognized for its comprehensiveness and reliability of the indexed publications. A list of search strings was formulated to conduct the initial search in the database, which consisted of two clusters of key phrases. The first cluster included phrases of 'augmented reality' and its abbreviation 'AR'; the second cluster indicated the educational context, including phrases such as 'professional training' and 'professional education'. As a result, commonly used search strings included '*augmented reality* OR *AR* AND *professional training* OR *professional education*'. Our initial literature search yielded 3184 articles, which were downloaded electronically for the ensuing manual screening.

2.1.2. Manual Screening

At this stage, all selected articles were examined by the first and second authors based on the following three criteria: (1) Within the scope of AR in education—articles that describe virtual reality (VR) or mixed reality (MR) or those in non-educational fields were removed. (2) An exclusive focus on professional training context—articles in other educational contexts (e.g., K–12, higher education, special education) were excluded. (3) Peer-reviewed empirical studies—purely conceptual or review papers were not included in the main library. Additionally, the snowballing technique, also known as citation chaining, was employed in this stage to avoid missing any major literature. The manual screening process is depicted in Figure 1. After the manual screening, a total of 49 articles (see Appendix A) were selected to be included in the main library for analytic coding in the next stage.

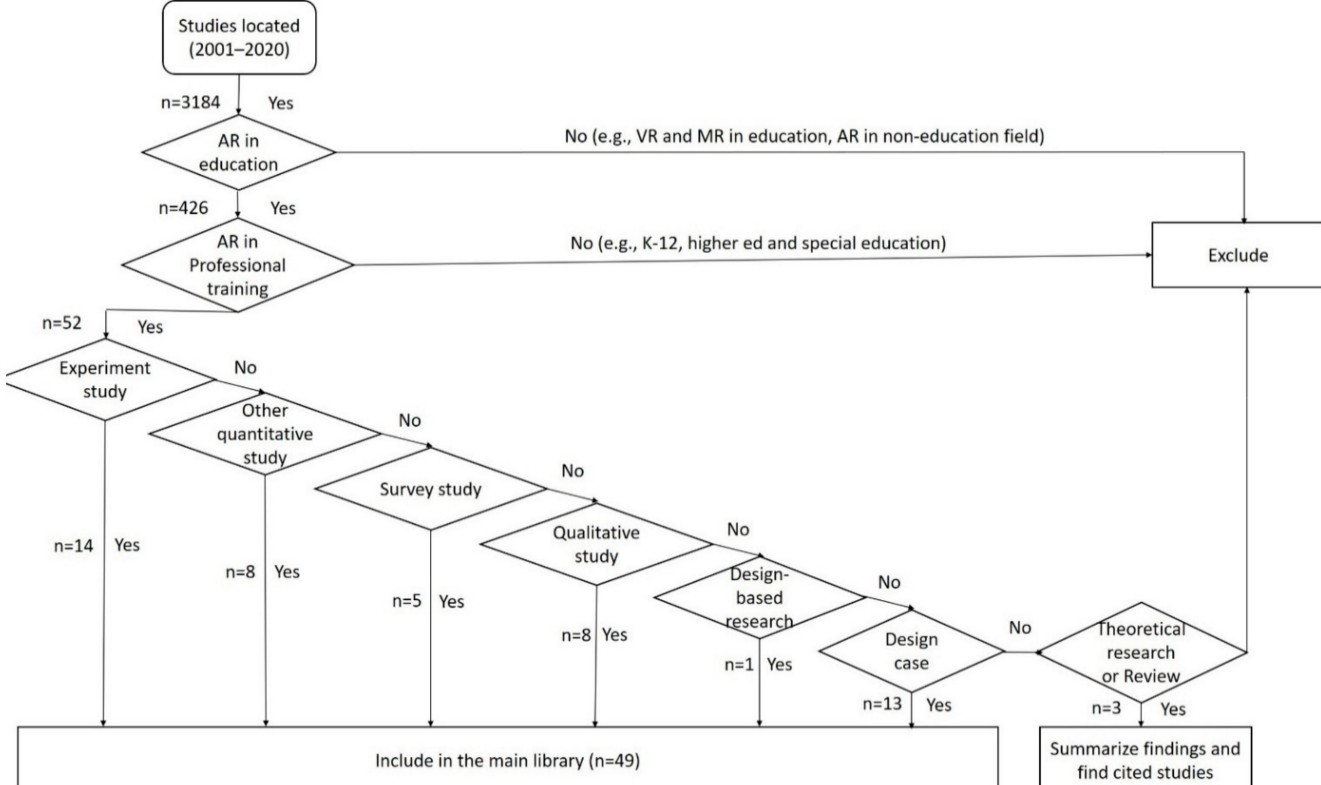

**Figure 1.** Decision diagram of the manual screening process.

### 2.2. The Data Coding and Analysis Processes

After finalizing the main library, we analyzed the content of the selected articles using the coding scheme as shown in Table 1. The coding scheme comprises five interrelated aspects regarding AR-supported professional training: basic information, instructional context, technology features, instructional design, and research results.

**Table 1.** Lists of codes for the analysis of selected articles.

| Category | Code | Description |
|---|---|---|
| Basic information | Title | Full title of the study |
| | Authors | Complete list of author names |
| | Year | Publication year |
| | Source | Information about the journal/book/URL |
| | Research type | Empirical/theoretical/synthesis |
| Instructional context | Disciplines | Engineering/health and medicine/other |
| | Implementation setting | Formal/informal |
| Technology features | Input | Voice/magnet/motion/haptic/GPS/mouse and keyboard/scanner/other |
| | Output | Monitor/video |
| | Computing devices | Desktop/laptop/mobile device/wearable device |
| | Media representation | Symbol/indicator/text/data/2D image/3D object/video/animation |
| | Interactivity | High level/low level/no interaction |
| Instructional design | Instructional function | Attention grabber/content delivery/practice/assessment/engagement/other |
| | Pedagogy | Game-based learning/trial-and-error/direct direction/experiential learning |
| | Scaffolding | No scaffolding/manual/computer |
| | Learning outcomes | Knowledge/behavior/skill/affective |
| Research results | Data source | Content Tests/surveys/interviews/videos/fieldnotes/other |
| | Statistical Results | Difference ($t$-test/ANOVA/MANOVA/ANCOVA/non-parametric), associational (SEM/regression/factor analysis), meta-analysis |
| | Effect size | Record if mentioned |

Information codes recorded the metadata of each article, including its publication year, title, authors, source, and research type. Context codes indicate the specific training fields in which AR was implemented, as well as the formality of implementation. Technology codes describe the technical features of AR interventions in the form of input, output, computing devices, media representation, and interactivity. Design codes specify the instructional functions of AR together with pedagogy and scaffolding design. Research codes are mainly concerned with the empirical findings, such as learning outcomes, data source, statistical results, and effect sizes of AR-supported professional training. The complete coding scheme and results are accessible at https://doi.org/10.17632/sd89r4zg56.2, accessed on 6 January 2022.

## 3. Results

This section may be divided by subheadings. It should provide a concise and precise description of the experimental results, their interpretation, as well as the experimental conclusions that can be drawn.

### 3.1. Publication Trends

Figure 2 reveals a small number of publications on AR-supported instruction in an educational context of professional training, with an overall upward trend during the past two decades. The results indicated that the application of AR in professional training did not gain much attention from researchers in the past; however, the prospects for the field were hopeful. As shown in Figure 2, there has been a significant increase since 2016, when Niantic and Nintendo launched Pokémon Go, a hugely popular location-based AR game. Within one week, the game had attracted over 65 million users and made the general public aware of this technological innovation [18]. As a result, the release of Pokémon

Go is believed to have promoted the commercialization and popularity of AR products. The maturity of AR technology and the media hype about it has led to its increased adoption and experimentation in the educational context. Based on AR development, we compared studies in two time periods: one from 2001 to 2015 and another from 2016 to 2020. Divided by the launching year of Pokémon Go, the two periods represent two phases of AR development and comprise relatively equal publication records.

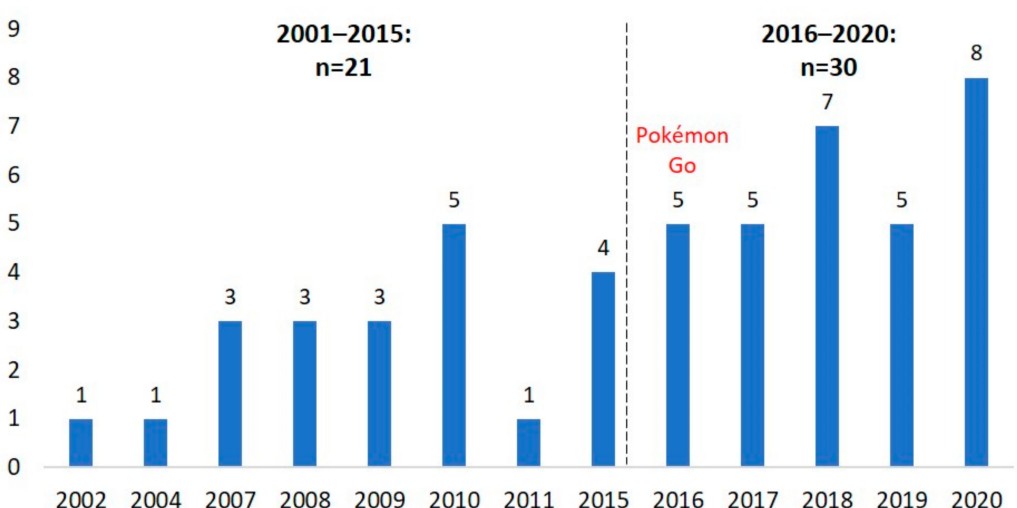

**Figure 2.** Publications on AR-supported instruction in professional training from 2001 to 2020.

Four categories of research studies were published from 2001 to 2020: theoretical research, empirical research, synthesis, and design case. As shown in Figure 3, the trend of publications in each research type is represented by colored lines. In this line chart, empirical research and design cases were the most common research types, but their evolving trends were different. The number of design cases increased before 2010 in general, reached the peak in 2010, and sharply decreased to zero in 2015. Regarding empirical research, there existed an overall rising trend, with a more significant increase occurring after 2011. Consequently, empirical research has been the mainstream research since 2011 to investigate AR usage in professional training. Compared with the two research types, theoretical research and synthesis were quite rare. Only one theoretical research was found in 2018, which discussed the impacts and application of AR on training in the age of e-learning [19]. Moreover, two synthesizing studies on AR in professional education were both published in 2020, respectively, reviewing applications of AR in organ transplantation [20] and research on AR used in sports education and training [21].

### 3.2. Instructional Contexts

From a total of 49 studies in professional education, AR-supported instruction was commonly used in two major fields: engineering and health and medicine. Twenty-five studies were in the field of health and medicine, accounting for more than a half, such as radiology, anatomy, and surgical operation. Fourteen studies were in the field of engineering, such as architectural engineering, automatic production, and machinery operations. In addition, the remaining 10 publications were in other fields, including basic science (e.g., biology, geography, agriculture technology) and social science (e.g., religious culture, teacher education). As the total number of these research studies was small, we combined them into the category of "Other". In terms of implementation settings, most research in the fields of engineering and health and medicine was conducted in formal settings, such as factories, workshops, and operating rooms. In contrast, studies in the

'Other' category were conducted in informal settings, such as campus sites, libraries, and general learning commons [22–24].

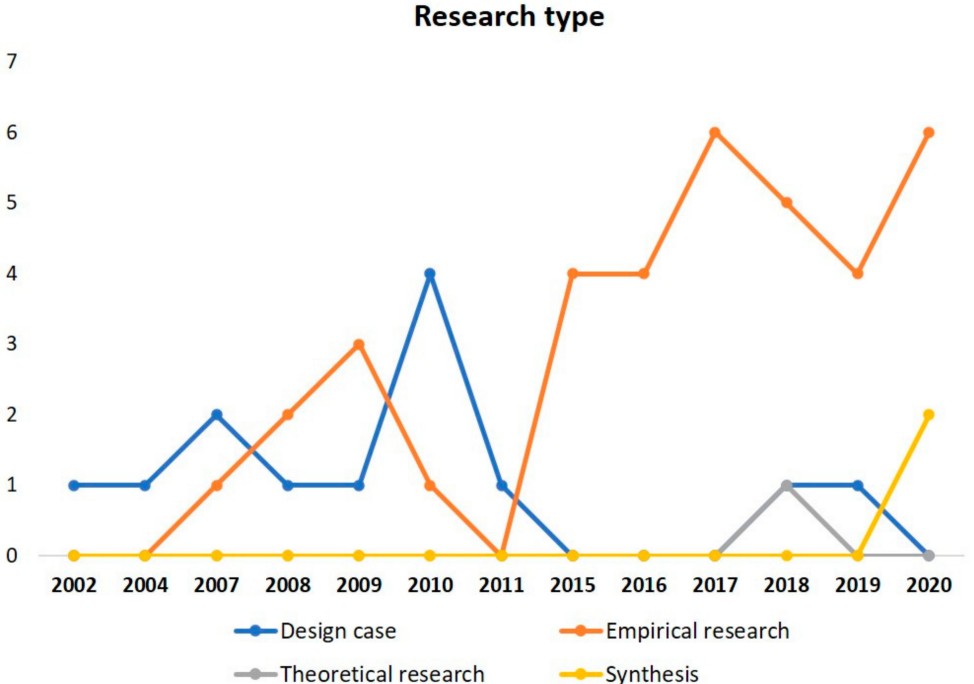

**Figure 3.** Research type of all AR-supported research in professional training from 2001 to 2020.

*3.3. Technological Features and Affordances*

### 3.3.1. Input and Output

User input is essential for interaction with AR interfaces, and the input devices can be classified into two major categories: natural input and artificial input. Natural input enables users to interact with AR objects in a more intuitive way, often without the reliance of accessory equipment. Common types of natural input devices include voice command, magnetic sensor (i.e., orienting behaviors such as tilt and rotation perceived by magnetic sensors), motion tracker (i.e., gestures, gait, and motions perceived by motion sensors such as Kinect), and haptic sensor (i.e., haptic stimuli synthesized using force feedback or tactile devices). Contrarily, artificial input relies on the use of peripheral devices to achieve unnatural interaction with AR systems. Commonly used artificial input devices include global positioning system (GPS) (e.g., location-based mobile applications), mouse and keyboard, scanner (e.g., image or QR code reader), and button-pressing controller (e.g., gamepad or joystick).

As seen in Figure 4, natural interaction was applied more frequently in professional training in the first 15 years, especially motion trackers (27%) and haptic sensors (19%). For example, the AR-supported simulator in surgical operation or ultrasonic examination was usually equipped with motion sensors to capture the instrument's motion and control users' operation [25–27]. In addition, users' haptic stimuli such as pressing or touching were traced by haptic sensors [28,29]. However, in the last five years, natural inputs seemed to lose their appeal to professional educators as their proportion dwindled substantially. In particular, the use of motion trackers witnessed a sharp decrease from 27% to 10%.

Contrarily, the recent literature has reported more instances of AR program featured by natural input. For example, scanning significantly increased (from 16% to 27%), accounting for a quarter of the research in professional training. In this way, AR devices were equipped with a camera or webcam to scan the QR code to help students to acquire instructional content, such as texts, images, and videos [30,31]. Another artificial type, inputting by mouse and keyboard, also showed an increase since 2016 (from 5% to 12%). The reduction in motion-based input and the increase in artificial types suggest the technical

difficulty in integrating motion trackers into AR and the need to further reduce the cost of AR development.

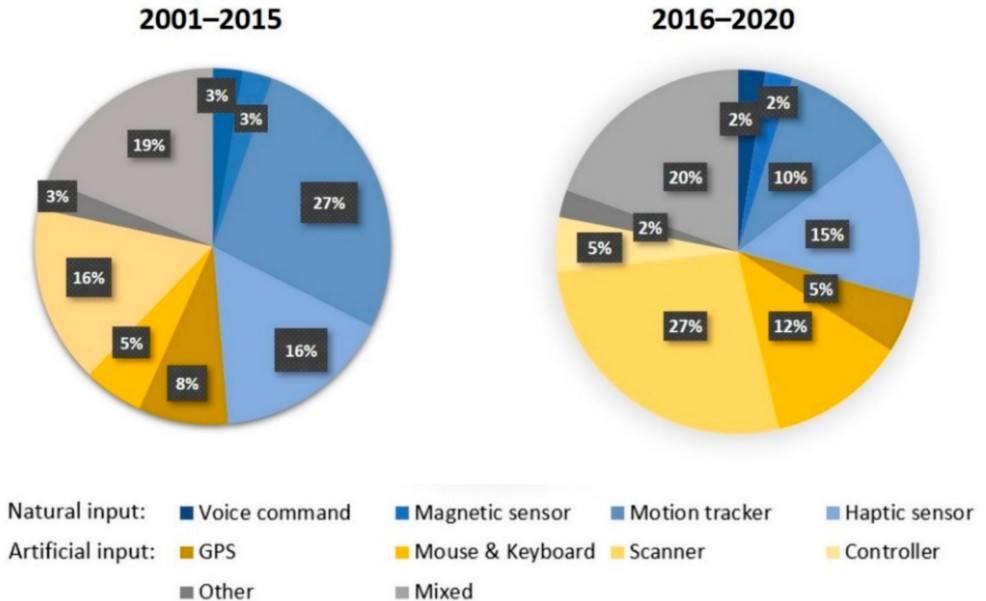

**Figure 4.** Input devices of AR-supported instruction from 2001 to 2020.

Compared with the types of input devices, output devices were simple, including three common types: output through monitor, video, and optical output. The former two types were used mostly in professional training, in which the combination of real images and virtual scenes was output in the form of video to certain display devices (e.g., mobile phones, tablets, HMD, glasses). The optical output type, depending on less artificial devices, was used in K–16 education much more than in professional training [28,32,33].

### 3.3.2. Computing Devices

In professional training, data in AR systems were processed by three main types of devices: the desktop or laptop, mobile devices (including tablets, phones, and other handheld devices), and wearable devices. Figure 5 shows a decreasing trend in desktop and an increase in mobile devices, consistent with technological advancements. In the first 15 years in the 21st century, the desktop/laptop had an overwhelming advantage over other computing devices, owing to the function of performing complex mathematical operations, satisfying the needs of simulation and calculation of AR systems. Meanwhile, in the last five years, portable devices, such as mobile or wearable devices, gradually replaced desktops and were used in some creative AR programs. For instance, Phan and Choo (2010) introduced an AR system in architectural education. For the convenience of virtual architecture representation, a set of wearable devices was designed for learners so that they could freely move in the scenario, which was combined with virtual images and real outdoor scenes, so as to help learners deeply understand the structure of the architecture and accept training [34].

### 3.3.3. Media Representation

In AR-supported instruction, the information carrier received by users had seven main forms, as shown in Table 2. In terms of the total number, 3D object was the most common media representation, especially in health and medicine. For example, in anatomical education and dental morphology, AR was used to construct a 3D model of the skull or dental piece for students to learn about the structure and composition [35,36]. In addition, video and text were two other important information carriers. In surgical training, the real-time changes in the operating environment were represented in videos for monitoring,

and trainees' operating processes were also recorded by videos [37,38]. Furthermore, text was used as a scaffolding to assist students to complete training or learning in AR systems in engineering and other fields [22,30,39]. Moreover, data occurred in health and medicine, recording the operating time, smoothness of the AR simulator, and the needle position [25].

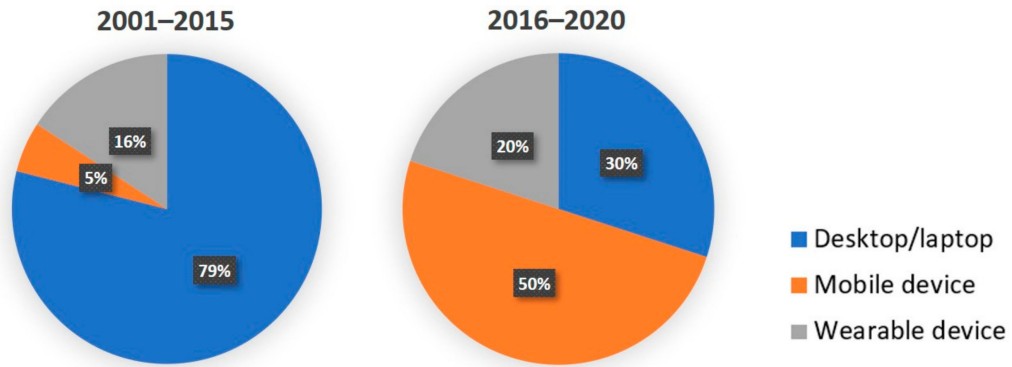

**Figure 5.** Computing devices of AR-supported instruction in the periods of 2001–2015 and 2016–2020.

**Table 2.** Media presentation of AR-supported instruction in three disciplines.

| | Symbol/ Indicator | Text | Data | 2D Image | 3D Object | Video | Animation | Mixed | Total |
|---|---|---|---|---|---|---|---|---|---|
| >Engineering | 1 | 4 | 0 | 3 | 8 | 5 | 4 | 4 | 29 |
| Health and medicine | 2 | 3 | 3 | 4 | 16 | 6 | 2 | 8 | 44 |
| Other | 1 | 4 | 0 | 2 | 3 | 1 | 0 | 4 | 15 |
| Total | 4 | 11 | 3 | 9 | 27 | 12 | 6 | 16 | 88 |

### 3.4. Instructional Design

#### 3.4.1. Pedagogy

In order to ensure the effectiveness of AR technology in professional training, it is essential to understand the pedagogy in AR intervention [40]. Over the past two decades, four pedagogies were found to be used in three main disciplines of professional design; however, their frequency of applying was different owing to instructional contexts and teaching content. Much of the research was design case, not applicable to the pedagogy, thus the number of papers in each category was less than the total number, especially in the field of engineering.

As seen in Figure 6, AR technology mainly served two pedagogies: trial-and-error (*n* = 13) and experiential learning (*n* = 12), which were consistent with the employment-oriented feature of professional training. Trial-and-error was defined as a process of repeated attempts with or without improvements by learning from failures [41]. It was found to be the most used pedagogy in health and medicine and was in line with the features of medical education, which required repeated operation practice to improve behavioral and practical achievement. For example, in health and medicine, residents used the AR simulator to operate based on the checklist. In this process, the instructor assigned the tasks, observed students' operating behaviors, and recorded their operating time and accuracy [14,42,43].

Another pedagogy that was used frequently, experiential learning, argued that knowledge is created and required through the transformation of experience, and knowledge results from the combination of grasping and transforming experience [44]. Compared with trial-and-error, experiential learning focused on not only learners' behavioral performance and skills, but the cognitive aspect of professional training, such as learning by doing, transfer learning, and reflection [19]. In AR-supported professional training, learners interacted with the AR system, then completed a questionnaire about the acceptance

and satisfaction of AR intervention, and even participated in an interview to reflect their learning experience [22,30,45].

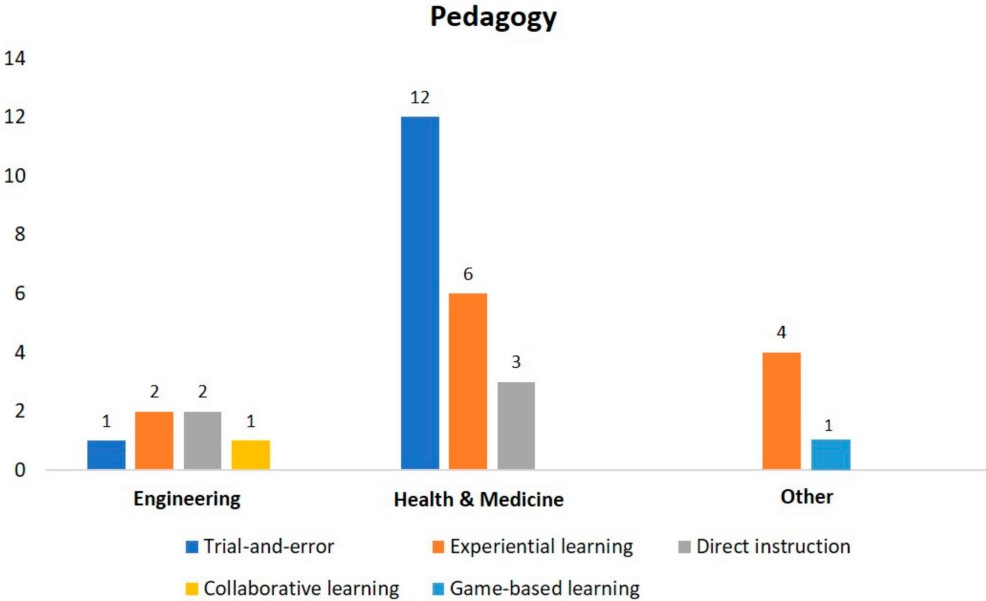

**Figure 6.** Pedagogies guiding AR-supported instruction in three disciplines.

### 3.4.2. Types of Learning Outcomes

Figure 7 reveals learning outcomes by discipline in the perspectives of knowledge, behavior, and affection. In engineering and health and medicine, students' behaviors/skills received the most attention (*n* = 23), such as trainees' operating time, accuracy, and proficiency in the workshop or simulated operating table [33,37,46–49]. In fact, this result depended on the characteristic of engineering and medical education, which emphasized practical operation skills. Moreover, some research focused on students' knowledge level (*n* = 8), which was measured by traditional exams before or after AR-supported instruction, to test the degree to which students had mastered new knowledge [35,50–52]. Another important finding is that affective outcomes (*n* = 11) were also noted in selected papers, such as their learning experience, motivation, and attitude to the AR intervention [30,53–55]. These affective achievements were commonly measured by questionnaires, interviews, or scales. Moreover, some papers focused on multiple learning outcomes simultaneously (*n* = 6), mixing their knowledge, behaviors, and affection, focusing on students' comprehensive development [36,52,56–58].

### 3.4.3. Instructional Function and Interactivity

The instructional function of AR intervention was divided into six types, among which content delivery and practicing accounted for most percentages and indicated significant changes in the 20 years. As seen in Figure 8a, the function of practice was far more than other functions in the first 15 years, consistent with the learning outcome of behaviors. Since 2016, practicing function decreased and the function of content delivery had a large growth and reached the top. In fact, the instructional function trend could be explained by the change in interactivity (Figure 8b). In the first 15 years, the AR system was of a high level of interactivity, so it could support student training and practice. From 2016 to 2020, the AR system in professional training improved substantially in technological development and could realize more educational functions. For example, a complete AR system could be provided as a representation tool to present and simulate something invisible to assist content delivery [35,36,52,58]. In this case, the main function was not to promote practice, so the degree of interactivity was lower.

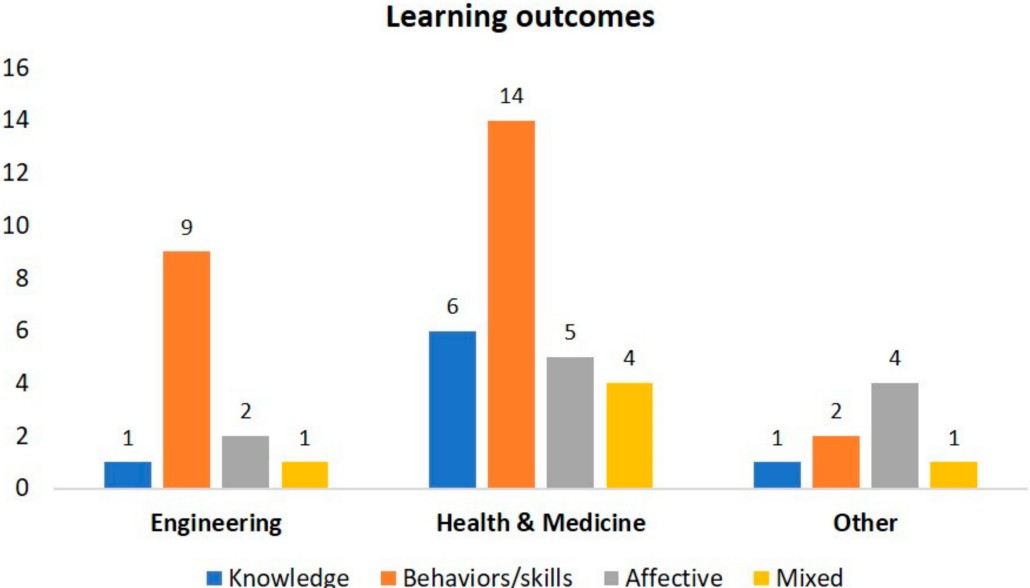

**Figure 7.** Learning outcomes in AR-supported instruction in three disciplines.

### 3.5. Meta-Analysis

To understand the overall effectiveness of AR-supported instruction and its moderating factors, we conducted a meta-analysis of the experimental research papers from 2001 to 2020. There were a total of 36 experimental research articles, but only 6 articles had reported necessary statistical information such as the mean, standard deviation, and sample size, meeting the requirements of the meta-analysis. The 6 articles included 15 separate experimental studies based on different dependent variables. The standardized effect sizes, as measured by Hedge's *g* [59], were calculated using the random-effects model (REM). The effect size ranged from $-2.498$ to $3.303$. Among all the empirical studies, nine were positive effects, five were insignificantly negative effects, and one showed no effect. The REM results revealed an overall small effect size of AR-supported instruction on learning outcomes ($g = 0.268$, $SE = 0.338$, $CI = [-0.395, 0.931]$, $p = 0.428$).

Furthermore, we conducted a mixed-effect analysis (MEA) to explore the potential moderators influencing the effectiveness of AR-supported instruction, and the results are indicated in Table 3. In order to avoid statistical heterogeneity and biased results, those with only one paper were not included in the calculation [60]. We examined eight potential moderators, and the MEA results revealed that there were seven variables that significantly moderated the effectiveness of AR-supported professional training. First, in terms of the training context, the results showed that discipline significantly moderated the effectiveness of AR-supported professional training ($Q_B = 6.43$, $p < 0.05$), with a large positive effect in engineering ($g = 1.748$) and an insignificantly negative effect in health and medicine.

From the perspective of research design, the results revealed significant variance in effect size in terms of pedagogy ($Q_B = 6.843$, $p < 0.05$) and instructional function ($Q_B = 13.555$, $p < 0.05$), while effect size did not vary according to the scaffolding mode ($Q_B = 2.551$, $p > 0.05$). Regarding pedagogy, the mixed pedagogy was the most effective ($g = 1.748$) for AR-supported training, while trial-and-error generated an insignificantly negative effect ($g = -0.246$). In addition, instructional function, both content delivery ($g = 1.204$) and mixed function ($g = 1.748$), reached a large effect size; however, the results should be carefully interpreted for only showing two and three effect sizes. However, the effect size of the practice function was insignificantly negative, consistent with the effect size of trial-and-error pedagogy.

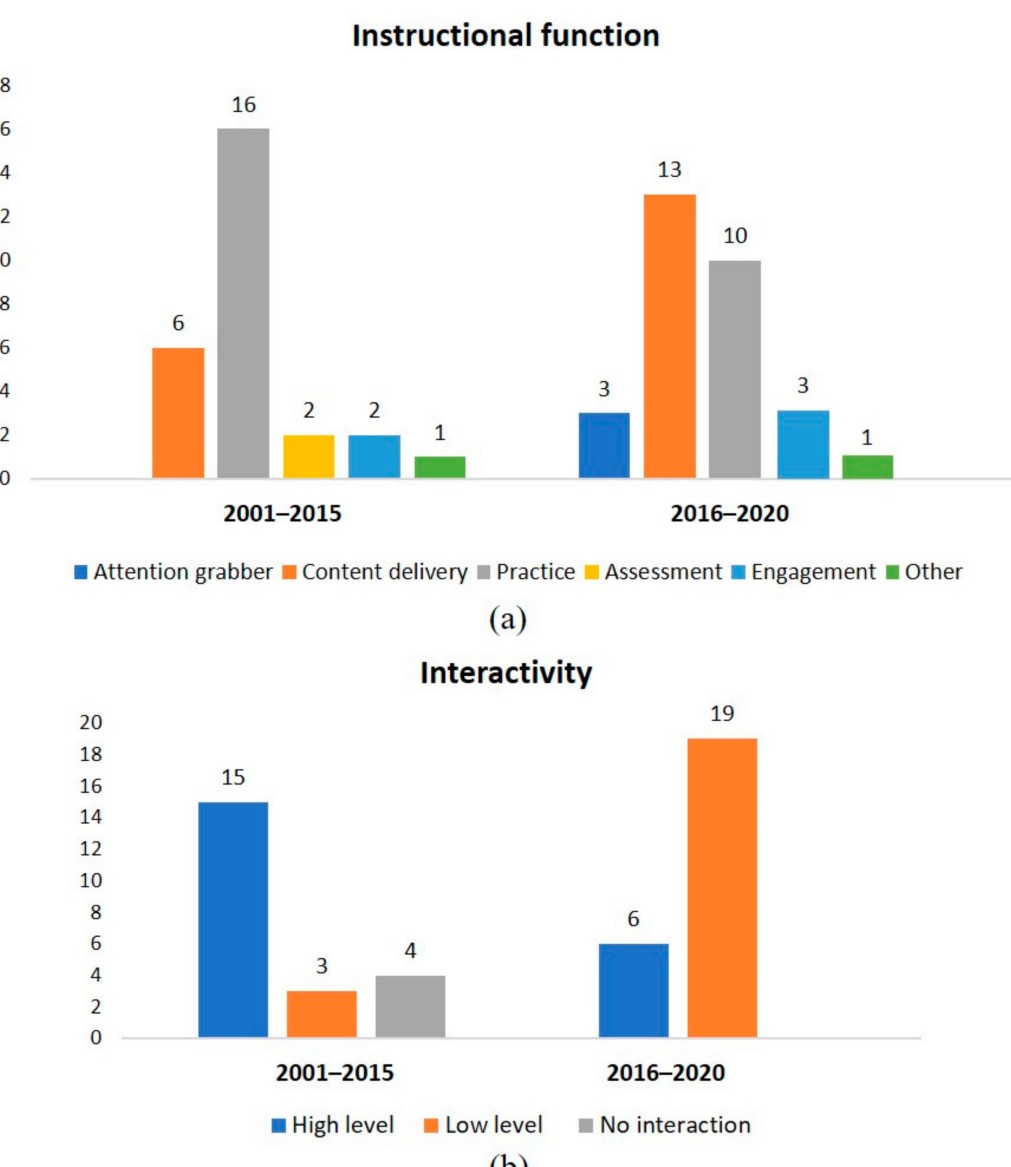

**Figure 8.** (**a**) Instructional function of AR-supported instruction in two periods of 2001–2015 and 2016–2020; (**b**) AR interventions by level of interactivity.

**Table 3.** Moderator analysis of selected experimental studies.

| Moderator | K | *g* | 95% *CI* | $Q_B$ | *p*-Value |
|---|---|---|---|---|---|
| Discipline | | | | 6.43 ** | 0.009 |
| Engineering | 3 | 1.748 | [0.421, 3.075] | | |
| Health and medicine | 11 | −0.246 | [−0.932, 0.441] | | |
| Pedagogy | | | | 6.843 ** | 0.009 |
| Mixed | 3 | 1.748 | [0.421, 3.075] | | |
| Trial-and-error | 11 | −0.246 | [−0.932, 0.441] | | |
| Instructional function | | | | 13.555 ** | 0.001 |
| Content delivery | 2 | 1.204 | [0.625, 1.783] | | |
| Mixed | 3 | 1.748 | [0.421, 3.075] | | |
| Practice | 10 | −0.359 | [−1.082, 0.364] | | |

**Table 3.** *Cont.*

| Moderator | K | $g$ | 95% CI | $Q_B$ | $p$-Value |
|---|---|---|---|---|---|
| Scaffolding | | | | 2.551 | 0.279 |
| No scaffolding | 8 | 0.346 | [−0.876, 1.568] | | |
| Computer | 3 | 0.520 | [−0.120, 1.160] | | |
| Manual | 4 | −0.079 | [−0.485, 0.327] | | |
| Input | | | | 12.080 ** | 0.007 |
| Controller | 4 | −0.996 | [−2.557, 0.564] | | |
| Scanner | 3 | 1.748 | [0.421, 3.075] | | |
| Haptic sensor | 4 | 0.749 | [0.104, 1.395] | | |
| Mixed | 4 | −0.0749 | [−0.485, 0.327] | | |
| Computing devices | | | | 11.721 ** | 0.003 |
| Desktop/laptop | 8 | −0.547 | [−1.414, 0.320] | | |
| Mobile devices | 4 | 1.666 | [0.742, 2.590] | | |
| Not mentioned | 3 | 0.520 | [−0.120, 1.160] | | |
| Output | | | | 14.609 ** | 0.001 |
| Monitor-based | 7 | −0.755 | [−1.636, 0.126] | | |
| Video see-through | 5 | 1.506 | [0.752, 2.259] | | |
| Not mentioned | 3 | 0.520 | [−0.120, 1.160] | | |
| Media representation | | | | 19.417 *** | 0.000 |
| 3D object | 6 | −0.539 | [−1.640, 0.562] | | |
| Text | 3 | 1.748 | [0.421, 3.075] | | |
| Video | 4 | −0.079 | [−0.485, 0.327] | | |
| Mixed | 2 | 1.204 | [0.625, 1.783] | | |

** $p < 0.01$; *** $p < 0.001$.

Regarding the technology of AR intervention, the Q-statistics indicated significant differences in the effectiveness of AR-supported professional training with devices of input modes ($Q_B$ = 12.080, $p < 0.05$), computing devices ($Q_B$ = 11.721, $p$ = 0.003 < 0.05), output modes ($Q_B$ = 14.609, $p$ = 0.001), and media representations ($Q_B$ = 19.417, $p < 0.001$). First, compared with the other input mode, scanning ($g$ = 1.748) input was most effective; haptic stimuli came second ($g$ = 0.749), while controller and mixed mode generated insignificantly negative effects. In terms of computing devices, mobile devices were the most effective on AR-supported instruction ($g$ = 1.666), while desktop/laptop yielded an insignificantly negative effect ($g$ = −0.547). Regarding output, the results revealed that video produced the largest effect size ($g$ = 1.506) and, interestingly, monitor-based output generated an insignificantly negative effect ($g$ = −0.755) despite its popularity (k = 7). Furthermore, in terms of media representation, text ($g$ = 1.748) and mixed ($g$ = 1.204) media representation yielded large effect sizes, while 3D object ($g$ = −0.539) and video ($g$ = −0.079) exerted insignificantly negative effects on AR-supported professional training.

## 4. Discussion and Conclusions

Based on the review results of 49 selected articles regarding AR-supported professional training, we provided tentative answers to the four research questions presented. First, the publication of AR-supported professional training has witnessed an overall upward trend since 2001, with design cases and empirical studies being the dominant research types before and after 2015. However, most studies were implemented in the contexts of engineering and health and medicine. Second, the advancement of AR technology has led to increased portability of its computing devices, yet the intuition of human–computer interaction suffered a slight setback, with artificial approaches (e.g., scanner, mouse, and keyboard) outweighing natural approaches (e.g., motion trackers) in terms of input in the last five years. Third, two widely used pedagogies for AR-supported instruction turned out to be trial-and-error and experiential learning, which are effective approaches for practice-oriented professional training. Moreover, content delivery has emerged as the main instructional function of AR for professional training in recent years. Lastly,

the meta-analysis results indicated that AR had an overall small effect on professional training outcomes, and the factors of discipline, pedagogy, instructional function, input, output, computing devices, and media representation moderate the effectiveness to varying degrees. Informed by the review results, practical implications and future research agenda are discussed below.

### 4.1. Implications for Practice

From the perspectives of technical function and instructional design, we propose the following implication for designing and implementing AR-supported professional training. Turning first to the technical functions of AR, the human–computer interaction mechanism can be further improved to allow more natural input for AR so that the perceived authenticity and ease of use can be further enhanced. Furthermore, we recommend increased wearable equipment to be used as a computing device of AR for improved portability and flexibility, allowing trainees to freely move in workplaces in order to operate and practice without restraint. Lastly, the co-presence of multiple media representations is a preferable AR display for professional training. Particularly, digital text overlay proves to be highly effective when used alone or in combination with other media types, despite its relatively low development cost.

In terms of instructional design, instructors should not be limited to one single pedagogy when facilitating AR-supported professional training. Despite the popularity of trial-and-error and experiential learning, flexible usage of multiple strategies seems to yield the best instructional effect. Moreover, we recommend using AR mainly for content delivery rather than hands-on practice in professional training. While many researchers have explored the instructional function of AR for practice, its effect has been unsubstantial. One possible reason is that the current AR functionality cannot afford authentic practice simulation, and the development of highly interactive AR for practice activity would further increase training costs.

### 4.2. Implications for Future Research

Based on our results, we proposed several implications for future research to address the limitation of current literature. First, the relatively small body of publications identified in this review suggests the need for more empirical investigations in this area. In addition to impact studies, a variety of research methodologies should be utilized to provide answers to more practical questions such as why AR works, in what contexts, for which population, and how.

Second, the scope of research on AR-supported professional training should be further expanded. Other training contexts other than engineering or health and medicine deserve further investigation, such as the service sector, manufacturing industry, and teacher development. Replicate studies in different contexts can further boost the credibility and generalizability of the research findings.

Third, the complete summary statistics required by meta-analysis should be properly reported in future research studies to allow the calculation of AR's aggregated effect on professional training. Several experimental studies reviewed in this study were missing key statistics such as means and standard deviation, which highlights the urgency to address this research limitation.

Lastly, we encourage conducting a cost analysis of AR-supported professional training in addition to determining its effect size. As argued by Kraft (2020), effect size alone does not reflect the cost of a program, and thus cannot justify its value for large-scale implementation [61]. Therefore, it is necessary to examine the cost-effectiveness of AR programs in order to make rational decisions regarding their adoption in professional training contexts.

**Author Contributions:** Conceptualization, X.H. and H.L.; methodology, Q.F. and H.L.; validation, Q.F. and H.L.; formal analysis, X.H. and Y.C.; writing—original draft preparation, X.H.; writing—review and editing, Y.C., Q.F. and H.L.; visualization, X.H.; supervision, H.L.; project administration, H.L.; funding acquisition, H.L. All authors have read and agreed to the published version of the manuscript.

**Funding:** This research was funded by the National Natural Science Foundation of China grant number 62177021.

**Institutional Review Board Statement:** Not applicable.

**Informed Consent Statement:** Not applicable.

**Data Availability Statement:** The data presented in this study are openly available in Mendeley Data at https://doi.org/10.17632/sd89r4zg56.2 (accessed on 5 January 2022).

**Conflicts of Interest:** The authors declare no conflict of interest.

## Appendix A  Research Articles Selected for the Systematic Review

| Author and Year | Article Title | Research Type | Discipline | Doi |
|---|---|---|---|---|
| Gelenbe, E. (2002) | Simulating autonomous agents with augmented reality | Design case | | 10.1016/j.jss.2004.01.016 |
| Weidenbach, M. (2004) | Intelligent training system integrated in an echocardiography simulator | Design case | Health and Medicine | 10.1016/S0010-4825(03)00084-2 |
| Lacey, G. (2007) | Mixed-reality simulation of minimally invasive surgeries | Design case | Health and Medicine | |
| Magee, D. (2007) | An augmented reality simulator for ultrasound guided needle placement training | Empirical research | Health and Medicine | 10.1007/s11517-007-0231-9 |
| Wang, X. (2007) | Design, strategies, and issues towards an Augmented Reality-based construction training platform | Design case | Engineering | |
| Botden, S.M.B.I. (2008) | ProMIS augmented reality training of laparoscopic procedures face validity | Empirical research | Health and Medicine | 10.1097/SIH.0b013e3181659e91 |
| Feifer, A. (2008) | Hybrid Augmented Reality Simulator: Preliminary Construct Validation of Laparoscopic Smoothness in a Urology Residency Program | Empirical research | Health and Medicine | 10.1016/j.juro.2008.06.042 |
| Koehring, A. (2008) | A Framework for Interactive Visualization of Digital Medical Images | Design case | Health and Medicine | 10.1089/lap.2007.0240 |
| Anastassova, M. (2009) | Automotive technicians' training as a community-of-practice: Implications for the design of an augmented reality teaching aid | Empirical research | Engineering | 10.1016/j.apergo.2008.06.008 |
| Botden, S.M.B.I. (2009) | Suturing training in augmented reality: Gaining proficiency in suturing skills faster | Empirical research | Health and Medicine | 10.1007/s00464-008-0240-2 |
| Harders, M. (2009) | Calibration, registration, and synchronization for high precision augmented reality haptics | Design case | Health and Medicine | 10.1109/TVCG.2008.63 |
| Chimienti, V. (2010) | Guidelines for implementing augmented reality procedures in assisting assembly operations | Design case | Engineering | 10.1007/978-3-642-11598-1_20 |
| Leblanc, F. (2010) | Hand-assisted laparoscopic sigmoid colectomy skills acquisition: Augmented reality simulator versus human cadaver training models | Empirical research | Health and Medicine | 10.1016/j.jsurg.2010.06.004 |
| Phan, V.T. (2010) | Developing outdoor augmented reality for architecture representation in educational activities | Design case | Engineering | |

| Author and Year | Article Title | Research Type | Discipline | Doi |
|---|---|---|---|---|
| Watanuki, K. (2010) | Augmented reality-based training system for metal casting | Design case | Engineering | 10.1007/s12206-009-1175-9 |
| Zhang, J. (2010) | A multi-regional computation scheme in an AR-assisted in situ CNC simulation environment | Design case | Engineering | 10.1016/j.cad.2010.06.007 |
| Behzadan, A. H. (2011) | A colllaborative augmented-reality-based modeling environment for construction enginerring and management education | Design case | Engineering | 10.1109/WSC.2011.6148051 |
| Abhari, K. (2015) | Training for planning tumour resection: Augmented reality and human factors | Empirical research | Health and Medicine | 10.1109/TBME.2014.2385874 |
| Chowriappa, A. (2015) | Augmented-reality-based skills training for robot-assisted urethrovesical anastomosis: A multi-institutional randomised controlled trial | Empirical research | Health and Medicine | 10.1111/bju.12704 |
| Cubillo, J. (2015) | Preparing augmented reality learning content should be easy: UNED ARLE-An authoring tool Empirical research for augmented reality learning environments | Empirical research | Health and Medicine | 10.1002/cae.21650 |
| Espejo-Trung, L.C. (2015) | Development and Application of a New Learning Object for Teaching Operative Dentistry Using Augmented Reality | Empirical research | Health and Medicine | 10.1002/j.0022-0337.2015.79.11.tb06033.x |
| Aivelo, T. (2016) | Digital gaming for evolutionary biology learning: The case study of parasite race, an augmented reality location-based game | Empirical research | Other | 10.31129/LUMAT.4.1.3 |
| Bourdel, N. (2016) | Augmented reality in gynecologic surgery: evaluation of potential benefits for myomectomy in an experimental uterine model | Empirical research | Health and Medicine | 10.1007/s00464-016-4932-8 |
| Chin, K.-Y. (2016) | Development of a mobile augmented reality system to facilitate real-world learning | Empirical research | Other | 10.1007/978-981-10-0539-8_36 |
| Juan, M.-C. (2016) | A mobile augmented reality system for the learning of dental morphology | Empirical research | Health and Medicine | |
| Reyes, A.M. (2016) | A mobile augmented reality system to support machinery operations in scholar environments | Empirical research | Engineering | |
| Díaz-Noguera, M.D. (2017) | Augmented reality applications attitude scale (ARAAS): Diagnosing the attitudes of future teachers | Empirical research | Other | 10.15804/tner.2017.50.4.17 |
| Moro, C. (2017) | The effectiveness of virtual and augmented reality in health sciences and medical anatomy | Empirical research | Health and Medicine | 10.1002/ase.1696 |
| Ozdamli, F. and Bal, E. (2017) | Pre-school teachers' views about educational materials and augmented reality in preschool education | Empirical research | Other | 10.21506/j.ponte.2017.8.35 |
| Ozdamli, F. and Hursen, C. (2017) | An Emerging Technology: Augmented Reality to Promote Learning | Empirical research | Engineering | 10.3991/ijet.v12.i11.7354 |
| Rochlen, L.R. (2017) | First-Person Point-of-View-Augmented Reality for Central Line Insertion Training: A Usability and Feasibility Study | Empirical research | Health and Medicine | 10.1097/SIH.0000000000000185 |
| Bacca, J. (2018) | Framework for designing motivational augmented reality applications in vocational education and training | Empirical research | Engineering | |

| Author and Year | Article Title | Research Type | Discipline | Doi |
| --- | --- | --- | --- | --- |
| Huang, C.Y. (2018) | The use of augmented reality glasses in central line simulation: "see one, simulate many, do one competently, and teach everyone" | Empirical research | Health and Medicine | 10.2147/ AMEP.S160704 |
| Indrawan, I.W.A. (2018) | Markerless augmented reality utilizing Gyroscope to Demonstrate the Position of Dewata Nawa Sanga | Empirical research | Other | 10.3991/ ijim.v12i1.7527 |
| Lee, D. (2018) | Augmented reality to localize individual organ in surgical procedure | Design case | Health and Medicine | 10.4258/ hir.2018.24.4.394 |
| Sirakaya, M. (2018) | Effects of augmented reality on student achievement and self-efficacy in vocational education and training | Empirical research | Engineering | 10.13152/ IJRVET.5.1.1 |
| Upadhyay, A.K. (2018) | In the age of e-learning: application and impact of augmented reality in training | Theoretical research | Other | 10.1108/DLO-04-2018-0041 |
| Zhu, E. (2018) | Understanding how to improve physicians' paradigms for prescribing antibiotics by using a conceptual design framework: A qualitative study | Empirical research | Health and Medicine | 10.1186/s12913-018-3657-x |
| Arts, E.E.A. (2019) | Face, Content, and Construct Validity of the Take-Home EoSim Augmented Reality Laparoscopy Simulator for Basic Laparoscopic Tasks | Empirical research | Health and Medicine | 10.1089/ lap.2019.0070 |
| Alismail, A. (2019) | Augmented reality glasses improve adherence to evidence-based intubation practice | Empirical research | Health and Medicine | |
| Kascak, J. (2019) | Implementation of Augmented Reality into the Training and Educational Process in Order to Support Spatial Perception in Technical Documentation | Design case | Engineering | 10.1109/ IEA.2019.8715120 |
| Lin, C.-H. (2019) | Research into the e-learning model of agriculture technology companies: Analysis by deep learning | Empirical research | Other | 10.3390/ agron- omy9020083 |
| Tzima, S. (2019) | Augmented reality applications in education: Teachers' point of view | Empirical research | Other | 10.3390/ educsci9020099 |
| Ashely-Welbeck, A. (2020) | Teachers' perceptions on using Augmented Reality for language learning in Primary Years Programme (PYP) education | Empirical research | Other | 10.3991/ ijet.v15i12.13499 |
| Boyaci, M.G. (2020) | Augmented Reality Supported Cervical Transpedicular Fixation on 3D-Printed Vertebrae Model: An Experimental Education Study | Empirical research | Health and Medicine | 10.5137/1019-5149.JTN.30733-20.2 |
| Coelho, G. (2020) | Augmented reality and physical hybrid model simulation for preoperative planning of metopic craniosynostosis surgery | Empirical research | Health and Medicine | 10.3171/ 2019.12.FO-CUS19854 |
| Ingrassia, P.L. (2020) | Augmented reality learning environment for basic life support and defibrillation training: Usability study | Empirical research | Health and Medicine | 10.2196/14910 |
| Kosieradzki, M. (2020) | Applicability of augmented reality in an organ transplantation | Synthesis | Health and Medicine | 10.12659/ AOT.923597 |
| Lester, S. (2020) | Some pedagogical observations on using augmented reality in a vocational practicum | Empirical research | Engineering | 10.1111/bjet.12901 |
| Soltani, P. (2020) | Augmented reality tools for sports education and training | Synthesis | Other | 10.1016/j.compedu. 2020.103923 |
| Xiao, J. (2020) | Assessing the effectiveness of the augmented reality courseware for starry sky exploration | Empirical research | Other | 10.4018/ IJDET.2020010102 |

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
