# Peer review of "Augmented Reality in Professional Training: A Review of the Literature from 2001 to 2020"

_applsci, doi:10.3390/app12031024_

Round 1
Reviewer 1 Report
AR research training from 2001 to 2020 is well organized. AR meta analysis is useful to understand the recent trend of the AR research with p-values. Selection and category of the AR research is well organized. However, authors need to use professional English service or ask native English colleague professors because there are some broken English grammars in entire manuscript. Therefore, manuscript can be minor revision.
- Between line 17 and 18, there are empty space.
- Please provide feature application or remove that.
- Figure 1 fonts are not clear.
- Authors need to use abbreviated journal names in the reference section.
- Figure 4 fonts need to be increased.
- In Table 1, authors can increase the space of the Table with raw direction.
- Description of the instruction input in Figure 4 need to be in detail.
- In Figure 5, there is "mixed". What is mixed ?
Reviewer 2 Report
The paper has a comprehensive review of the augmented reality technology used for professional training.
The manuscript has a traditional structure (Introduction, Methods, Results, Discussion & Conclusions) and is well-presented.
Please pay attention to the following comments:
- The 2nd paragraph of the paper (lines 40-50) explains “professional training”. This part should be expanded. I recommend indicating the specific areas of professional training to which the study is directed.
- The presented decision diagram is clear. But in my opinion, the scope ‘augmented reality OR AR AND professional training OR professional education’ used for literature search was narrow. Major papers were missed.
- The described references didn’t present all the experience of using Augmented Reality for professional training. Particularly, the studies in training using AR for (1) lean manufacturing classes, (2) production management classes, (3) improving the spatial skills in working with 2D drawings, (4) supporting spatial perception in technical documentation, (5) for the realization of assembly procedures by workers, (6) for supporting the process of material picking by workers, etc.
- Do not duplicate phrases from the title in the Keywords. It is suggested to use other terms/phrases to expand the visibility of your paper in the network.
